# A Systematic Review of Community-Based Exercise Interventions for Adults with Intellectual Disabilities

**DOI:** 10.3390/healthcare13030299

**Published:** 2025-02-01

**Authors:** Teresa Greene, Laurence Taggart, Gavin Breslin

**Affiliations:** 1School of Nursing and Midwifery, Queen’s University Belfast, Belfast BT9 7BL, UK; laurence.taggart@qub.ac.uk; 2School of Psychology, Queen’s University Belfast, Belfast BT9 5BN, UK; g.breslin@qub.ac.uk

**Keywords:** intellectual disability, physical activity, physical activity level, exercise, exercise interventions, community, community based, systematic review

## Abstract

**Background**: Adults with intellectual disabilities are a particularly inactive sub-population who experience disproportionally poorer health and social exclusion when compared with the wider general population. This systematic review aimed to identify whether community-based exercise interventions were theoretically underpinned, whether they had an active single- or multi-exercise component, and how the interventions were objectively and/or subjectively measured, as well as deducing if they improved the health of this population. **Method**: A systematic search of five databases was conducted up to May 2024. The study was registered in PROSPERO and followed PRISMA reporting guidelines. Study methodological quality was appraised using the Critical Appraisal Skills Programme. Risk of bias was determined using the Cochrane collaboration tools ROB 2 and ROB 2 CRT. Articles were eligible for inclusion if they recruited adults with intellectual disability, were community-based, had an active exercise component and measured physical activity (PA) levels as an outcome measure. **Results**: In total, 9034 records were identified, with five studies meeting the eligibility criteria. All studies used a feasibility RCT or RCT methodology and all focused on weight loss or PA levels as the primary outcome measure. Two studies focused on walking as a single exercise, two combined walking with nutrition/weight loss, and one combined walking with aerobics. One multi-component walking and aerobics intervention led to statistically significant improvements in PA. Multi-component community-based exercise interventions led to statistically significant improvements in body composition measures. No studies showed statistically significant improvements in quality of life. Risk of bias was rated moderate to high across all included studies. **Conclusions**: Caution should be taken in drawing firm conclusions due to the small number of included studies, small sample sizes and high risk of bias. Multi-component community-based interventions are more effective at improving PA levels in adults with intellectual disabilities than walking-only studies. Future studies should be theoretically underpinned and explore the use of peer and student models of social support. The use of fitness facilities such as leisure centres and gyms requires further exploration in this population.

## 1. Introduction

Despite the vast benefits of physical activity (PA) for physical and mental health, just 9% of adults with intellectual disabilities engage in PA levels which meet the World Health Organisation minimum recommended guidelines of 150 min of moderate intensity per week in bouts of at least 10 min [1,2,3]. The need for increased involvement in exercise programmes that promote PA levels leading to health benefits are arguably even more pertinent in adults with intellectual disabilities who commonly present with complex health profiles. These include, but are not limited to, high rates of cardiovascular disease, diabetes, osteoporosis, and mobility problems [4,5,6]. There is growing evidence suggesting that these major health conditions may be alleviated by managing negative lifestyle risk factors such as poor diets, high levels of sedentary behaviour, and low levels of physical activity leading to high levels of obesity [4,6,7,8]. This is alongside many adults with intellectual disabilities being prescribed high levels of anti-psychotropic medication leading further to abdominal obesity [9].

Prevailing health inequalities experienced by adults with intellectual disabilities also contribute to their sustained social exclusion [10]. Social inclusion is a core domain of quality of life with many benefits to people with intellectual disabilities [11,12], yet this population remain socially excluded and require support from various agencies to support full societal inclusion [13,14]. Increased encounters with members of the public, especially repeated over time and involving a shared interest, could encourage more community participation amongst adults with intellectual disabilities [15]. One method of increasing such encounters, with the added benefit of health improvement, is through community-based PA interventions. A scoping review of initiatives to promote social inclusion in people with intellectual disabilities found that community-based PA interventions promoted positive attitudes towards people with intellectual disabilities [16].

The evidence for the benefits of community-based exercise in the general population is well developed and highlights the benefits of the community setting in contributing to a range of improvements beyond physical measures. Systematic reviews have identified links between the social benefits in community settings in PA interventions to increased exercise adherence and improved self-esteem [17,18]. A one-year longitudinal study of older adults participating in social groups which included physical activities found social wellbeing to be improved in the community setting, and loneliness significantly reduced [19]. A meta-analysis examining the factors which increase motivation for PA found that interventions in community-based gym and fitness settings delivered by fitness professionals in those settings had larger effects on theoretical components such as stages of change and multiple motivational constructs [20].

There is some evidence to suggest the benefits of PA within community settings for people with intellectual disabilities. Participation in PA can lead to increased social connectedness in adolescents and young adults with intellectual disabilities [21]; considering the context and environment within which exercise occurs could be important for also enhancing social inclusion. Exercising in nature (or ‘green exercise’) may provide greater satisfaction or enjoyment than indoor activities [22] as well as providing an opportunity for engagement in community-based exercise. Indoor community options for exercise including public gyms and fitness centres could afford people with intellectual disabilities benefits that green spaces do not. Indoor fitness facilities can provide a broad range of cardiovascular and strength training activities within more secure and environmentally predictable location [23,24].

Existing systematic reviews of the effectiveness of PA interventions for adults with intellectual disabilities have resulted in largely inconclusive and inconsistent results [25,26]. Brooker et al. [25] noted a dearth of robust PA interventions and the need to utilise more valid and reliable outcome measures for this population. In their systematic review, Hassan et al. [26] concluded that there were serious concerns about the fidelity of how such PA interventions for adults with intellectual disabilities were delivered across settings. Hassan and colleagues also reported that the majority of studies in their review had no positive effects on PA and reiterated the need for rigour in the research designs.

Despite these findings, effectiveness has been identified in community-based interventions that utilise transactional research methodologies, where research is translated into evidence-based practices in real-world settings [27,28]. The limited effectiveness of interventions that promote PA in this population may be attributed to a lack of understanding of the contexts within which they occur [21]. One systematic review has focused on the recruitment settings and delivery contexts of health promotion programmes for adolescents and young adults with intellectual disabilities [29], whilst another has considered community factors for PA promotion from a socio-ecological perspective [30], but neither have explored the impact of the community setting specifically. This is the first systematic review to synthesise the evidence for PA-promoting community-based interventions for adults with intellectual disabilities.

The concept of ‘community’ has not been universally defined in the literature, and can refer to a feeling of belonging amongst groups of people with similar values [31]. People with intellectual disabilities are more likely to socialise, work, and live with other people with disabilities, and whilst this is not to be negated, better models of ‘genuine’ inclusion should be worked upon [31]. For the purposes of this review, the term ‘community-based’ refers to interventions which have taken place in environments which are also accessed by members of the general public.

The aim of this systematic review is to synthesise the evidence for community-based exercise interventions for adults with intellectual disabilities to ascertain the following: (1) What community-based exercise interventions are there for adults with intellectual disabilities? (2) Are these community-based exercise interventions theoretically underpinned? (3) What objective and subjective outcome measures do these community-based exercise interventions use? (4) Do these community-based exercise interventions improve the physical, mental, and/or social wellbeing of adults with intellectual disabilities?

## 2. Materials and Methods

This review has been registered on the international prospective register of systematic reviews (PROSPERO: CRD42023440659). The Preferred Reporting Items for Systematic reviews and Meta-Analyses (PRISMA) guidelines [32] were used in the design and reporting of this review.

### 2.1. Search Strategy

A systematic search of peer-reviewed studies was conducted in five databases: Ovid MEDLINE, Ovid EMBASE, Cumulative Index to Nursing and Allied Health Literature (CINAHL), PsycINFO, and the Cochrane Central Register of Controlled Trials. Searches comprised keywords relating to ‘intellectual disability’ and its synonyms combined with keywords relating to ‘exercise’ and ‘physical activity’. MeSH terms, truncation, and wildcard functionality were used where appropriate. Initial searches were conducted in June 2023 and were updated in May 2024. Studies were limited to publication in or after 1995, the English language, and adult only (aged 18+).

### 2.2. Eligibility Criteria

For inclusion in this review, studies were required to have (1) a study population which was exclusively adults with intellectual disabilities, (2) with a pre–post-intervention study design (3), which was a community-based exercise intervention, (4) contained a structured PA component which was repetitive and dosed, and (5) measured PA as an outcome measure. In this review, ‘community-based’ interventions were defined as interventions which took place in locations which were used by other members of the public such as gyms, leisure centres, and/or public parks. Studies were limited to those that were full-text, peer-reviewed, and in English only.

### 2.3. Exclusion Criteria

Studies that exclusively recruited individuals with Prader–Willi syndrome were excluded as interventions specified to meet the unique needs of this population were not likely to be relevant for the wider intellectual disability population. Interventions which recommended exercise or consisted of information giving only were not included. Studies that described interventions which took place in private rooms of community facilities did not meet the author’s criteria of community-based, along with university/laboratory style settings or the participant’s place of work or residence.

### 2.4. Data Management and Screening

Titles were imported into reference management software EndNote 21 (EndNote, Clarivate, Philadelphia, PA, USA) where duplicates were deleted. Titles were considered at the abstract level then reviewed at the full-text level for inclusion using a screening tool based upon the pre-defined inclusion and exclusion criteria, as per the study protocol (available on request). The reference lists of the titles which were retained at the full-text level were also hand searched and screened. Potentially relevant studies were reviewed by the first author independently on the basis of the title and abstract. Following this, potentially relevant titles were screened independently at the full-text level by the first and third authors (TG, GB) using the aforementioned screening tool. Queries or disagreements on inclusion were resolved through discussion and consensus in collaboration with the second author (LT).

### 2.5. Data Extraction

Relevant and detailed outcome data were independently extracted from the included studies using a structured tool developed by the research team on Microsoft Excel (Version 16.89.1 Microsoft Corp., Redmond, WA, USA) (available on request) by two authors (LT and GB) and cross-checked by the primary author for consensus. Extracted data included author and study title, study aims, study design, duration of study, theoretical framework, funding source for study, conflicts of interest, participant characteristics (number of participants, gender, mean age, diagnosis/level of disability, inclusion and exclusion criteria of the study, co-morbidities or other general health problems, baseline differences between control and intervention groups, attrition rates), method of recruitment, adverse events, description of intervention and control group, outcomes measured, methods used to measure outcomes, time points measured, results, statistical methods used, and key conclusions. Extracted data were summarised and tabulated.

### 2.6. Quality Appraisal and Risk of Bias

Study methodological quality was appraised using the Critical Appraisal Skills Programme [33], which appraises study quality based on validity and generalisability. As there are no specific CASP checklists for non-randomised pilot or pre–post-test quantitative studies, the checklist for RCT was used for all studies. Risk of bias was examined and graded using the Cochrane collaboration’s range of tools for different intervention types: risk of bias tool for randomised trials (RoB 2) and cluster randomised trials (RoB 2 CRT) [34,35]. Quality appraisal of the studies was completed by the first and second authors, with differences resolved by the third author if necessary.

### 2.7. Outcome of Interest

The primary outcome of interest in this review is the effect of community-based exercise interventions on the PA levels of adults with intellectual disabilities. The overall mean changes in PA levels from baseline to endpoint will be considered by objective (pedometers, accelerometers) or subjective (self-report) means.

### 2.8. Data Analysis and Synthesis

Due to the dearth of identified studies in this review and their heterogeneity, results are presented narratively. A formal narrative synthesis was conducted and is reported in line with SWiM guidelines [36]. Studies are grouped for synthesis by intervention design (methodology, setting, sampling, recruitment strategy) and by outcomes (the method of physical activity measurement and any other reported outcome measures of physical, psychological or social wellbeing).

## 3. Results

The database searches generated 8944 titles, with 6912 remaining after the deletion of duplicates. A flow chart detailing the reasons for exclusion is presented in Figure 1. A total of 73 studies were retained for full text review. The studies were reviewed at the full-text level by the first and third authors. A total of five studies met the full inclusion criteria. Due to the great degree of heterogeneity between the studies in design, delivery, and outcomes, a meta-analysis was not possible, and the findings are reported descriptively. An overview of the studies and their participant characteristics (Table 1) and the outcomes measured in the included studies (Table 2) are presented.

### 3.1. Community-Based Exercise Interventions for Adults with Intellectual Disabilities

An overview of the included studies and participant demographics are presented in Table 1 and Table 2, respectively. One study utilised a randomised controlled trial design [37], two used a cluster randomised control trial design [38,39], and two used a feasibility randomised trial design [40,41]. The included studies were based in Australia, Scotland, and the USA.

Three of the studies were walking interventions [39,40,41]. Two studies were multi-component weight loss interventions that primarily focused on nutrition but also recommended walking as a source of PA to supplement weight loss [37,38]. As walking was the recommended method of PA in all the studies, the location where PA took place was mostly outdoor green and urbans spaces.

Sampling and recruitment methods were similar across the included studies, with studies adopting a convenience sampling approach or using targeted recruitment, which focused on providers of intellectual disability services. Sample sizes ranged from 16 to 150 participants, and across all five studies, the total number of participants was 352. The mean age of the participants recruited to the studies ranged from 21.4 to 44.9 years old. Two studies recruited adults with severe and profound intellectual disability [38,39], whilst the other three included adults with mild or moderate intellectual disabilities [37,40,41]. One study exclusively recruited adults with Down syndrome [41], and one exclusively recruited adults with autism spectrum disorder and intellectual disabilities [40]. Reasons for excluding participants were also similar across all the included studies, but only two studies reported the presence of co-morbidities in their participant sample in detail [38,39].

**Table 1 healthcare-13-00299-t001:** Overview of included studies.

	Shields & Taylor [41]	Melville et al. [39]	Ptomey et al. [37]	Savage et al. [40]	Harris et al. [38]
Location	Melbourne, Australia	Glasgow, Scotland	Kansas, USA	Two unspecified US states	Glasgow, Scotland
Study methodology	Phase II randomised trial (pilot study)	Cluster randomised controlled trial	Randomised controlled trial	Feasibility RCT	Cluster randomised controlled trial
Theoretical framework	Rimmer and Roland conceptual intervention model [42]	Social cognitive theory, transtheoretical model	Social cognitive theory	N/A	N/A
Study description	Walkabout study. A walking intervention compared with a social programme control group.	Walk Well study. A walking intervention compared with a waiting-list control group.	Enhanced stop light diet vs. conventional diet. A comparison of two dietary approaches.	Step It Up Study. Comparison of a supported self-managed exercise programme with access to FitBit devices only.	TAKE5 vs. Waist Winners Too. Comparison of a multi-component weight management programme (TAKE5) with a health education programme (WWToo).
Intervention group	Walkabout group—participants were paired with a student mentor to complete two 45 min walking sessions per week. Participants were also encouraged to complete an additional 60 min of walking per week to achieve a total of 150 min/week of moderate intensity activity.	Walk Well—participants aimed to gradually increase their daily walking time to 30 min (3000 steps) on at least 5 days of the week by Week 12. Participants had three PA consultations where they set individualised goals with a walking advisor and a carer.	Enhanced stop light diet (eSLD)—a diet consisting of −2 portion-controlled entrees per day, 2 portion-controlled shakes per day, 5 servings of fruits and vegetables per day and additional meals, snacks, and calorie-free drinks featured in the eSLD guide. Participants were advised to achieve 150 min/week of PA. Brisk walking was recommended.	Step It Up—participants were given a FitBit and FitBit resources and were supported by a coach (family or paid carer) to increase their PA. Participants scheduled 2 days per week to focus on walking and aerobic activities that would increase their step count for 30 min per session.	TAKE 5—participants attended TAKE5 sessions 1–2 times per month approximately. It included a personalised diet plan with a deficit of 600 kcal/day and individualised walking goals which progressively increased.
Control group	Social activities—participants were paired with a student mentor to engage in social activities that would not have a training effect once a week for 90 min.	Waiting list control group—usual care	Conventional diet (CD)—participants were encouraged to achieve a 5–700 kcal/day energy deficit and were provided with information to meet energy intake goals. Participants were given the same PA guidance as the eSLD group.	Access to a FitBit and FitBit resources only—participants were given the device and instructions on how to wear it, sync it and use it to monitor their step count.	WWToo—health education programme which was delivered 1–2 times per month. Participants were given advice on nutrition. PA goals were discussed and reviewed at sessions.
Setting	Local community—walking	Local community—walking	Local community—walking	Local community—walking or local fitness facilities	Local community—walking or local leisure facilities and clubs
Duration	8 weeks	12 weeks	6 months weight loss phase 12 months maintenance phase	12 weeks	6 months weight loss phase 12 months maintenance phase
Support source	Undergraduate physiotherapy students	Study walking advisors Family and/or paid carers	Family and/or paid carers	Family and/or paid carers or other paid professional	Dietician and health professional—programme delivery Family and/or paid carers
Sampling method and recruitment strategy	Convenience sample of individuals who had participated in previous studies conducted by the research team.	Targeted approach using a multi-point strategy, recruiting individuals from a range of intellectual disability providers.	Targeted approach was used by identifying disability organisations within a 50-mile radius of Kansas City.	Targeted approach via autism groups in two US states.	Multi-point recruitment strategy—recruiting from specialist intellectual disability services, provider organisations, and local day centres.

**Table 2 healthcare-13-00299-t002:** Participant demographics, retention, and compliance rates in the included studies.

	Shields & Taylor [41]	Melville et al. [39]	Ptomey et al. [37]	Savage et al. [40]	Harris et al. [38]
No. of participants	16	102	150	34	50
Mean age (years)	21.4 ± 3.2	44.9 ± 13.5 Intervention 47.7 ± 12.3 Control	36.1 ± 12.0 Intervention 37.0 ± 12.5 Control	23.75 ± 5.67 Intervention 29.72 ± 10.85 Control	40.6 ± 15.0 Intervention 43.6 ± 14.0 Control
Diagnosis/level of disability	Mild and moderate intellectual disability, all with Down syndrome	N = 48 mild N = 25 moderate N = 8 severe intellectual disability	N = 26 Down syndrome N = 20 autism N = 103 other All had mild-moderate intellectual disability	All had diagnosis of mild or moderate intellectual disability Mean IQ scores for both groups Intervention—60.65 ± 10.23 control—62.69 ± 7.67	N = 14 mild ID N = 21 moderate ID N = 5 severe ID N = 7 profound ID N = 8 Down syndrome
Co-morbidities	None stated	Epilepsy (9.8%) Visual impairment (54.9%) Hearing impairment (19.6%) Mental ill health (32.3%) Problem behaviours (17.6%)	Intervention—49.4% prescribed obesogenic medication Control—45.8% prescribed obesogenic medication	Medication usage N = 22 (64.7%) Self-reported motor skills: N = 13 poor (38%) N = 19 typical (55.8%) N = 2 advanced (5.8%)	Epilepsy N = 11 (22%) Vision impairment N = 25 (50%) Hearing impairment N = 9 (18%) Mental health problems N = 9 (18%) Problem behaviour N = 19 (38%) High blood pressure N = 23 (46%) Obesogenic medication N = 15 (30%) Type 2 diabetes N = 4 (8%)
Retention (%)	2 months—100%	3 months—79.6%	6 months—83.3% 18 months—65.3%	3 months—95%	6 months—90%
Compliance	Exercise sessions attended Intervention—123/128 (96%) Control—63/64 (98%)	PA consultations completed (/3) 3–71% 2–26% 1–3%	Attendance at monthly meetings Intervention—80% Control—76% Dietary recommendations at 6 months 18 months Entrees—9%, 10% Shakes—17%, 0% Fruit and veg—28%, 31%	Procedural fidelity—90.9%	Attendance at >75% of programme sessions Intervention—19/24 (79%) Control—16/24 (66%)

Study duration ranged from 8 weeks [41] to 6 months with additional 12 month maintenance phases [37,38]. Participant retention rates across all included studies were good, ranging from 65.3% [37] to 100% [41]. The lowest retention rates were seen in the study of longest duration, where 65.3% were retained after 18 months [37]. Compliance to intervention components was measured in a number of different ways across studies. Most commonly, compliance was measured as the number of scheduled check-in or exercise sessions attended. Compliance and fidelity measures across the studies in relation to attendance at sessions or adherence to study protocols were generally good at between 66 and 90.9%. One study reported compliance to their dietary recommendations as very poor, ranging from 0 to 31% [37]. In four studies, family members and paid carers were either encouraged or formally recruited to provide support to participants for the study duration [37,38,39,40]. Two of these studies also utilised other individuals to provide support; dieticians and health professionals delivered the weight management programmes in Harris et al. [38], whilst walking advisors provided advice and support in Melville et al. [39]. One study recruited undergraduate physiotherapy students as mentors to exercise with participants for the duration of the intervention [41].

### 3.2. Theoretical Underpinnings of the Community-Based Exercise Interventions

Social cognitive theory [43] provided the theoretical underpinning for two of the studies [37,39], one of which also used the transtheoretical model of behaviour change [39]. One study based their intervention on a conceptual intervention model by Rimmer and Roland [42], which included some constructs from both the transtheoretical model of behaviour change and social cognitive theory. Two studies did not refer to or use a theoretical model in programme development or evaluation [38,40].

### 3.3. Quality Appraisal

The CASP tool findings on overall study quality are outlined in Table 3. The use of numerical scoring of study quality is discouraged in the Cochrane handbook for systematic reviews of interventions [35], so study quality is expressed narratively.

### 3.4. Risk of Bias

Risk of bias was examined and graded using the Cochrane collaboration’s range of tools for different intervention types: risk of bias tool for randomised trials (RoB 2) and cluster randomised trials (RoB 2 CRT) [34,35]. These risk of bias tools assess the risk of bias in relation to a particular outcome of interest rather than in relation to whole studies. In this review, the risk of bias was assessed in relation to physical activity levels. Risk of bias assessments were conducted independently by two authors (TG and LT) and agreed upon by consensus discussion with the third author (GB).

The Risk of bias VISualisation (Robvis) tool was used to assess the quality of the included studies across the risk of bias domains included in each tool. The Robvis tool contained the risk of bias judgements for each domain of the ROB2 (Figure 2) and ROB2 CRT tools (Figure 3). Summary graphs produced by the Robvis tool are available as Appendix A.

None of the studies scored a low risk of bias. Three studies were categorised as serious risk of bias [37,39,40] and two as moderate [38,41]. Two studies scored a high risk of bias due to missing outcome data [37,39]. In Ptomey et al. [37], PA data were reported descriptively as group averages rather than as means and standard deviations for each group. This also caused a high risk of bias in the reported results domain, as the failure to report means and standard deviations of PA data deviated from the published study protocol. In Melville et al. [39], baseline data were gathered for all consenting participants before randomisation, but the data for four clusters of participants who had withdrawn after this period was not reported. Savage et al. [40] was the only study to score a high risk of bias in the measurement out of the outcome domain, as outcome assessors were not blinded.

### 3.5. Objective and Subjective Outcome Measures

The included studies measured a mixture of physiological, psychological, and social outcomes, which are outlined in Table 4. The primary focus in two studies was weight management [37,38], and three studies primarily focused on the intervention’s effect on PA levels [3,39,40,41]. All of the studies measured PA levels objectively. Accelerometers were used in four studies and were the most frequently used method of recording PA data, while two studies additionally used pedometers to record step counts [39,41], and one used a FitBit Flex2 watch [40]. There was a great degree of heterogeneity between the studies in relation to the methods for collecting and reporting PA data. Minimum accelerometer wear time per day varied from 6 to 10 h, and epoch length ranged from 15 to 60 s. Minimum wear time criteria ranged from 6 to 10 h per day on 3 or 4 days of the week, while one study did not report wear time criteria [40]. Just one study used an 8-day data collection period to allow for the deletion of day 1 data to minimise potential wear effect bias [40].

All of the studies measured height and weight in order to calculate Body Mass Index (BMI), and four studies measured waist circumference (WC). One study measured physical fitness and walking speed [41]. Quality of life was the most frequently measured psychological outcome, measured in two studies with the European Quality of Life-5 dimensions (ED-5D) tool [38,39,40] and in one study [40] with the Quality of Life Questionnaire (Q.QOL) [44]. Two studies measured participants perceptions of wellbeing [39,41] using the Exercise Outcomes Scale [45] and Subjective Vitality Scale [46], respectively. One study measured life satisfaction [41] using the Life Satisfaction Scale [45], and one measured self-efficacy [39] using the Self-efficacy for Activity for Persons with Intellectual Disabilities [47]. Other outcome measures used in the studies in this review included walking speed, body composition, muscle endurance, muscle strength, flexibility, and life satisfaction.

**Table 4 healthcare-13-00299-t004:** Study outcomes as reported in the included studies.

	Shields and Taylor [41]	Melville et al. [39]	Ptomey et al. [37]	Savage et al. [40]	Harris et al. [38]
Method of PA measurement	RT3 accelerometer (Stayhealthy, Inc., Monrovia, CA, USA) Pedometer (Omron, Walking Style Pro, HJ-720ITE2)	Actigraph GT3X accelerometer (Manufacturing Technology Inc., Crestview, FL, USA) Omron Walking Style III pedometer (Omron Healthcare Inc., Hoffman Estates, IL, USA)	Actigraph GT1X accelerometer (Pensacola, FL, USA)	Fitbit Flex2	Actigraph GT3X+ accelerometers (ActiGraph, LLC, Pensacola, FL, USA)
Minimum wear time	10 h of data on at least 4 days out of 7 including 1 weekend day 8 days wear was used and day 1 data were not used due to wear effect	6 h of data on at least 3 days of the week out of 7 7 days wear total	8 h of data per day on at least 3 days out of 7 7 days of wear total	None stated Participants wore devices for 7 days	6 h of data on at least 3 days of the week out of 7 7 days wear total
Cut points	Not reported	Sedentary (<100) Time in PA (>100) MVPA (>1952 counts)	Troiano et al. [48] cut points MVPA (>2020 counts/min)	N/A	Sedentary (<100) Time in PA (>100) MVPA (>1952 counts)
Epoch	Not reported	15 s	60 s	N/A	15 s
Compliance	Logbook data cross-referenced with pedometer data	Number of physical activity consultation sessions attended	Attendance at monthly meetings, compliance to dietary plans, completion of self-monitoring plans and adherence to recommended PA guidelines	Fidelity was measured via weekly checklists, checking weekly goal setting via the Fitbit dashboard and via 4 video recorded goal setting meetings	Attendance in at least 75% of programme sessions
Fitness measure	6 MWT using 25 m course	X	X	X	X
Weight	Weighing scale (model not reported). Measure taken with shoes off.	SECA 877 scales (SE approval class III; SECA, Hamburg, Germany). Measured twice and mean value used.	Digital scale (Belfour model #PS6600, Saukville, WI, USA) Measured after over-night fasting in the morning.	Aria Wi-Fi Smart Scale	SECA877 scales (SE approval class III; SECA Germany). Measured twice and mean value used.
Height	Stadiometer (model not reported). Measurement taken with shoes off.	SECA Leicester stadiometer (SECA, Germany). Measured twice and mean value used	Portable stadiometer (#Invicta Plastics Limited, model IP0955, Leicester, UK)	Measured, equipment type not specified.	SECA Leicester stadiometer (SECA, Germany). Measured twice and mean value used
Waist circumference (WC)	Two measures of WC taken and a third taken if the first two measures disagreed by more than 0.3 cm. No detail on how measures were taken.	Measured at mid-point between the iliac crest and the lowest rib, in full expiration with the participant standing. Measured twice and mean value used.	Lohman et al. [49] procedure used. The average of the closest two of three measurements was recorded.	X	Measured at mid-point between the iliac crest and the lowest rib, in full expiration with the participant standing. Measured twice and mean value used.
Walking speed	GAITrite system	X	X	X	X
PSYCHOSOCIAL SCALES					
Perceptions of wellbeing	Exercise Outcomes Scale [45]	Subjective Vitality Scale [46]	X	X	X
Life satisfaction	The Life Satisfaction Scale [45]	X	X	X	X
Self-efficacy	X	Self-efficacy for Activity for Persons with ID [47]	X	X	
Quality of life	X	European Quality of Life-5 dimensions [41]	X	Quality of Life Questionnaire (QOL.Q) [44]	The European Quality of Life-5 dimensions (EQ-5D) youth version

MVPA—moderate to vigorous physical activity; ID—intellectual disability; 6 MWT—six-minute walk test.

### 3.6. Physical, Mental, and/or Social Wellbeing Benefits of Community-Based Exercise Interventions

#### 3.6.1. PA Levels

PA data for each of the studies at each time point are detailed in Table 5. Due to differences in how PA data were collected and analysed and to the heterogeneity of the methods of PA in the included studies, they are described descriptively.

Only one study [40] reported statistically significant improvements in PA levels. Average weekly counts increased from 47,420 steps to 60,241 in the intervention group; a statistically significant Time x Group ANOVA interaction effect was reported (*p* = 0.031). Control group participants had a small decrease in mean weekly steps from 46,227 (18,095) at baseline to 46,377 (6821) at post-intervention.

Three studies reported increases in PA which favoured the intervention group, though these were negligible and none showed statistical significance [38,39,41]. For example, in the Walkabout study [41], a small increase in PA levels in the intervention group between baseline (342.1 ± 108.8 vector counts/min) and post-intervention (344.2 ± 158.8 vector counts/min) was reported, where decreases were recorded in the control group. In this study, PA was also recorded through the use of self-report logbooks and pedometers, which indicated that an average of 146–175 min of walking per week was achieved by the participants, though baseline data were not reported so it is unclear if the intervention led to any changes. In the Walk Well study [39], a small 69.5 steps per day mean increase between the intervention and control group was observed by week 12, and a 79 steps per day mean increase for the intervention group from baseline to post-intervention. No increases were seen in MVPA levels, with participants in the intervention group at baseline achieving a mean of 3.2 ± 2.7 min/day of MVPA and 3.0 ± 2.6 min/day post-intervention at 12 weeks. Similarly, in the enhanced stop light intervention [37], mean MVPA levels in the intervention group decreased from 15 min/day for all participants at baseline and 12 min/day at 18 months.

#### 3.6.2. Physiological Outcome Measures

Weight was a recorded outcome measure in all of the included studies, and WC was also measured in four studies [37,38,39,41]. BMI was reported in three studies at post-intervention [37,39,40]. One study measured cardiovascular fitness [41].

Ptomey et al. [37] reported statistically significant reductions in WC (cm) (−5.2 ± 5.8, 1.8 ± 5.9, *p* = 0.001), weight (kg) (−6.8 ± 5.5 kg; −7.0%, −3.6 ± 5.3 kg; −3.8 kg, *p* = 0.001), and BMI (kg/m) (−2.4 ± 2.3, −1.4 ± 2.3, *p* = 0.015) after a 6-month weight loss period. Statistical significance of these differences was not maintained in the weight maintenance phase of the study (7–18 months). In Savage et al. [40], there was a statistically significant Time x Group interaction effect for weight, with a mean 3.25 pound weight loss at post-intervention for the intervention group and a mean increase in weight in the control group (*p* = 0.04). Harris et al. [38] reported statistically significant reductions in weight, BMI, WC, and body fat percentage in the intervention group at 6 months and 12 months. However, there were no statistically significant between-group differences at any time point in this study.

No statistically significant changes in weight, BMI, or WC were reported in two of the studies [39,41], but a moderate treatment effect which favoured the intervention group for WC and weight was observed in Shields and Taylor [41]. The intervention group decreased their mean WC from 95.6 ± 17.2 to 90.1 ± 12.1 cm at post-intervention with increases in the control group from 89.3 ± 8.8 to 94.1 ± 7.4. The intervention group lost a mean weight of 0.7 kg between baseline and post-intervention, and the control group gained a mean 0.3 kg by the post-intervention period.

One study measured cardiovascular fitness [41] using a six-minute walk test. Intervention group participants increased their mean walk distance by 36.8 m at post-intervention, and control group participants had a mean decrease of 8.2 m in their walk distance. Estimates of standardised mean difference (SMD) between groups indicated a moderate effect size which favoured the intervention group.

#### 3.6.3. Psychosocial Outcome Measures

Three studies measured quality of life [38,39,40], with one accepting the use of proxy respondents where participants had severe and profound intellectual disabilities [38]. Perceptions of wellbeing were measured in two studies [39,41], and one study measured self-efficacy [39]. All psychosocial outcome measures in these studies used tools which had been developed specifically for adults with intellectual disabilities, though it was not reported if they were reliability or validity tested. None of the studies reported statistically significant results for any of the psychosocial outcome measures at any time point.

Melville et al. [39] reported that the participants had difficulty understanding the measures used in their study, which impacts the validity and generalizability of the findings. Shields and Taylor [41] reported a small negative effect in the intervention group for the wellbeing measure from 14.7(2.1) at baseline to 14.3(3.4) at post-intervention. The study authors theorised that this may have indicated potential problems with the outcome measure used, or a small negative impact on wellbeing for the intervention group related to the effort required to engage in 150 min of moderate PA per week. Harris et al. [35] did not report any difficulties with quality of life measurement in their study, and were the only study who accepted the use of proxy responders such as family or paid carers.

None of the studies measured social inclusion as an outcome. Subsequently, the effect of community-based exercise interventions on the social inclusion of adults with intellectual disabilities remains unknown. The impact of the community-based environment was not referred to by the authors of any of the studies.

## 4. Discussion

This is the first systematic review of PA community-based exercise interventions for adults with intellectual disabilities. We aimed to identify whether these PA community-based exercise interventions were theoretically underpinned, had an active single- or multi-exercise component, and how the interventions were objectively and/or subjectively measured, as well as deducing if they improved the health of this population.

### 4.1. Theortical Underpinning

It is encouraging to note the use of theoretical models and their components in influencing the intervention designs of the identified studies in this review. Other systematic reviews relating to PA in intellectual disability populations have noted a distinct lack of reference to theoretical models and individual behaviour change theories in the design of their included studies [25,50]. In this review, two studies [37,39] were underpinned by social cognitive theory [43], including one [39] which combined this theory with the transtheoretical model of behaviour change [51]. Despite the inclusion of theoretical models in two of the included studies in this review, there was no evidence of the studies implementing behaviour change theories throughout the intervention process. The Walk Well study was the only study that measured a component of their chosen theoretical model (self-efficacy) as an outcome measure [39].

### 4.2. Single or Multi-Component

There were three single-exercise-component studies in this review which recommended only walking as a means to increase PA levels [37,39,41]. None of the single-component studies had statistically significant effects on PA, and only Ptomey et al. [37] reported statistically significant post-intervention results for other measures (WC, weight, and BMI). Two studies involved multi-component exercises by promoting walking along with additional activities which could contribute to increasing PA levels such as household tasks and accessing local fitness centres [38,40]. Multi-component interventions had more statistically significant results, with Savage et al. [40] reporting statistically significant increases in PA levels and decreases in weight at the post-intervention period. Harris et al. [38] reported statistically significant reductions in weight, BMI, WC, and body fat percentage in the intervention group at post-intervention, though no between-group differences reached statistical significance.

### 4.3. Objective/Subjective Measurement

The use of valid and reliable objective measures over both the studies was encouraging. Accelerometers were used in all of the included studies and provided objective, valid, and reliable data about PA levels and intensities. There were no reported difficulties from any of the studies in the use and meeting the wear time of the accelerometers as a data collection method, indicating their suitability for the intellectual disability population. The heterogeneity in accelerometer use amongst the included studies made the comparison of PA data between the studies difficult. Calls for a standardised protocol for accelerometer use in intellectual disability studies would result in higher-quality review data and allow for more accurate comparisons between studies [52].

A dearth of studies measured psychological outcomes, which is consistent with other systematic review findings [53]. Studies which included psychosocial measures of wellbeing commented on the difficulties participants had with comprehending questionnaire scales, which can be abstract in nature. While the practice of self-reporting should be encouraged over proxy respondents from an autonomy perspective, a review found that neither approach was preferable over the other [54]. In Harris et al. [38], proxy responders were used when appropriate to assist participants with severe intellectual disabilities to understand a quality of life scale. This effective use of proxy responders can ensure that those with more severe levels of intellectual disability can still be included in PA research.

### 4.4. Effectiveness of Interventions

Walking was the chosen PA component in all of the included studies, which were published in or after 2015, indicating recent trends in the literature towards this exercise type for this and other vulnerable populations. Two studies also encouraged participants to use other community-based resources to increase PA including leisure facilities, sports clubs, and exercise classes [38,40]. Walking as a method of improving PA levels is free and relatively accessible to most people with intellectual disabilities. Despite these advantages, walking was not a successful method of improving PA levels for adults with intellectual disabilities in the included studies in this review. Just one community-based exercise intervention including a combination of walking and attending the community gym was effective in significantly improving the PA levels of the adults with intellectual disabilities [40]. In the Shields and Taylor [41] pilot study, their walking intervention indicated feasibility due to high compliance rates, and overall, an average of 146–175 min/week walking was achieved by participants. However, these results should be interpretated with caution due to the studies’ pilot study design, moderate to high risk of bias, and small unpowered sample size.

Two of the studies in this review had a more positive impact on body composition measures, which is unsurprising when the predominant focus in these studies was dietary changes and weight loss, rather than PA promotion [37,38]. In these studies, the promotion of PA was less targeted, more passive and unstructured, and consisted of discussions with participants around what PA they could achieve on a weekly basis as part of a weight loss intervention.

Though the results for the included studies demonstrated little statistically significant effects on increasing PA, they provided opportunities for adults with intellectual disabilities to engage in community-based exercise where such opportunities are scarce [55]. Clinically important health effects have been seen in children with just a five-minute increase in PA, and WHO [2] PA guidelines stipulate that any improvements in PA are better than none. Good retention rates across all the included studies indicate that community-based exercise interventions are feasible for adults with intellectual disabilities. It was particularly encouraging to note that two studies recruited and retained participants who exceeded the power calculated target [37,39]. Recruitment and retention issues in clinical trials in cognitive disability populations are common [56], even in studies conducted in multiple countries where potential participant catchment areas are densely populated. There were no reported adverse events in any of the studies, and the positive attitudes of the participants to engaging in PA was noted by several of the included studies.

The community-based settings of the studies included in this review did not appear to translate into enhanced community participation for adults with intellectual disabilities. None of the studies in this review investigated the impact of the community setting on any outcomes relating to community inclusion or participation. The definition of ‘community-based’ in this review postulated that the setting where PA took place should be accessible to members of the general population and not be exclusively used by other people with intellectual disabilities. In a systematic review which included intellectual disability specific groups, such as Special Olympics, there was little evidence of their impact on improving community participation and inclusion for people with intellectual disabilities [55]. A scoping review of initiatives that aimed to facilitate social inclusion of people with intellectual disabilities in physical activities found that increasing awareness and inclusion of this population through physical activity programmes was possible [16], though long-term effectiveness remains unknown. A multi-modal approach to creating inclusive environments which takes into consideration accessibility of the physical environment, disability-positive policy and attitudes, and partnership between disability organisations and mainstream providers is needed [16].

### 4.5. Barriers and Facilititors to Implementation

Social support was identified by the majority of study authors as a strong influence on outcome measure results. In the Walk Well study [39], a subsequent process evaluation [57] reported that reliance on paid carers to support the participant to engage in walking was ineffective. Paid and family carers in this study were experiencing low morale, increased workloads, and time constraints, and paid carers had inconsistent contact with the study participants, which affected the motivation of family carers to engage fully with the intervention.

Studies where family carers were a predominant and reliable support source had better outcomes. Savage et al. [40] reported no issues with family carer support in their study, which was the only study in this review to achieve a statistically significant improvement in PA levels, and also reported high procedural fidelity (90.9% average). Harris et al. [38], in their subsequent process evaluation of the TAKE 5 intervention [58], commented on the strength of support from family carers who took on full responsibility for implementing the intervention, which resulted in statistically significant decreases in body composition outcomes for the participants. Shields and Taylor [41] utilised undergraduate physiotherapy students as a source of social support in their walking intervention for young adults with Down syndrome and reported high retention (100%) and compliance rates (96%) and positive PA outcome results. Future research in this area should place a targeted focus on robust support networks for adults with intellectual disabilities, especially where family or paid carers may struggle with additional workloads associated with research studies. More research into the use and sustainability of student and peer support models for adults with intellectual disabilities is needed given these preliminary yet encouraging examples of their effective use.

Contextual factors were described in one study as a barrier to walking outside, as participants perceived their local area to be unsafe, which prevented them from reaching PA goals [58]. Another study that included participants with severe intellectual disabilities found that behaviour change techniques, such as self-monitoring PA through pedometers and walking diaries, were too abstract and complex for participants and their carers to complete [39]. Savage et al. [40] did not recruit adults with severe intellectual disabilities and did not report any difficulties with participants understanding these concepts. The successful use of visual supports in Savage et al. [40] enabled participants to become more independent with exercising and goal setting.

### 4.6. Implications for Future Community PA Interventions

Future community-based PA interventions for adults with intellectual disabilities should draw upon the strengths of the studies included in this review. Though there are only five studies included in this review, they provide crucial information on both the enablers and challenges of developing, testing, and implementing interventions for this population. The use of valid and reliable objective data collection tools is a strength, and studies in this area should continue to use accelerometers with this population in order to contribute to the evidence on PA levels for adults with intellectual disabilities. This review has highlighted the lack of valid and reliable scales for measuring exercise attitudes, self-efficacy, and quality of life of adults with intellectual disabilities, which should be addressed in future studies.

The use of theory to underpin future interventions is crucial, and the use of individual behaviour change theories in the included studies is positive. However, two of the included studies did not test the components of their change theories in their outcome measures; therefore, the impact of the theory on the behaviour and attitudes of the participants is unknown. None of the included studies utilised a system change theory, despite the profoundly influential impact of environmental and contextual factors on the successful implementation of health-promoting interventions.

The included studies have shown that adults with intellectual disabilities have high rates of compliance in community-based exercise interventions, especially when given social support. Walking is cost-effective and within the physical capabilities of the vast majority of individuals with intellectual disabilities. One multi-component intervention resulted in a statistically significant increase in steps per week, indicating that walking in addition to increasing activity in home environments and community-based fitness environments can lead to statistically significant improvements in PA levels [40]. Increasing PA levels in community settings through walking-only interventions was not effective, and the sustainability of community-based walking programmes for adults with intellectual disabilities is also still unknown.

Developing new or adapting existing interventions for the intellectual disability population appears to be more complex and challenging than previously thought [56]. Particular attention should be given to the planning and preparation stages of future interventions. A particular focus should be given to the readiness of community settings to support people with intellectual disabilities in health-promoting PA interventions [59]. Dependence on paid carers may not be a viable option in the face of social care cuts and economic austerity, and dependence on family members who are at increased risk of physical health problems, stress, and burnout may also not be sustainable. The increased use of peer and student mentoring in effective health-promoting interventions for children and adults with intellectual disabilities is encouraging and should be explored further [41,60].

More interventions aimed at improving the physical health of individuals with intellectual disabilities is needed. Basing interventions in the communities where adults with intellectual disabilities reside may provide an opportunity to promote social inclusion, though no community-based exercise interventions for adults with intellectual disabilities have measured social inclusion as an outcome. Researchers developing PA interventions for adults with intellectual disabilities should consider basing them in community environments and measuring the potential benefits of the environment upon the social inclusion and quality of life of participants. All of the studies included in this review facilitated or recommended walking outdoors as a predominant source of PA enhancement, whilst just two recommended that participants could access other community amenities such as leisure centres or fitness classes. Indoor fitness facilities could provide a broader range of cardiovascular and strength training activities within more secure and environmentally predictable locations than outdoor green or urban spaces. Although the accessibility of these environments for people with disabilities remains an issue [23,24,61], if these barriers were overcome, more interaction with members of the general population in fitness spaces could translate to enhanced community inclusion, which intellectual disability specific settings cannot provide. Equal access to community-based amenities, including gyms and fitness suites, is a legal right for all adults with intellectual disabilities [62], yet use of these facilities is currently neglected in the literature. Whilst consideration should be made for overcoming barriers to fitness facilities for people with disabilities [63,64,65], these settings could provide a wider range and intensity of health-enhancing exercises than walking could provide [66].

### 4.7. Limitations of the Review

The ability to draw robust conclusions from this review is impeded by the low number of identified studies and their low levels of methodological quality. Meta-analysis of the data was not possible due to the heterogeneity of outcome measures used. Due to time and resource limitations, non-peer-reviewed literature was not included, and the initial searches and exclusions at the title and abstract level were conducted by one author (TG). Studies in a language other than English were excluded due to the unavailability of financial resources to fund translation, which may have unfortunately impeded the geographical scope of the included studies. Due to the participants recruited to the studies included in this review, older adults with intellectual disabilities, those with profound intellectual disabilities, those with behaviours of concern, and those from the continents of Asia, South America, and Africa are not represented.

## 5. Conclusions

Multi-component community-based interventions to increase PA levels in adults with intellectual disabilities are more effective than walking-only interventions. The utilisation of community-based gyms and fitness centres in PA-promoting interventions for this population is yet to be explored. Future research in this area should fully embed behaviour change theory and consider the utilisation of system change theories in their intervention design and outcomes. Social support is a strong facilitator for PA behaviour in adults with intellectual disabilities. Where consistent and reliable support from family and paid carers is not possible, peer support and undergraduate student sources of support could be effective alternatives. The psychosocial benefits of engaging in community-based PA exercise interventions are unknown in the intellectual disability population due to the dearth of studies measuring psychosocial outcomes. Future studies should include valid and reliable psychosocial measures.

## Figures and Tables

**Figure 1 healthcare-13-00299-f001:**
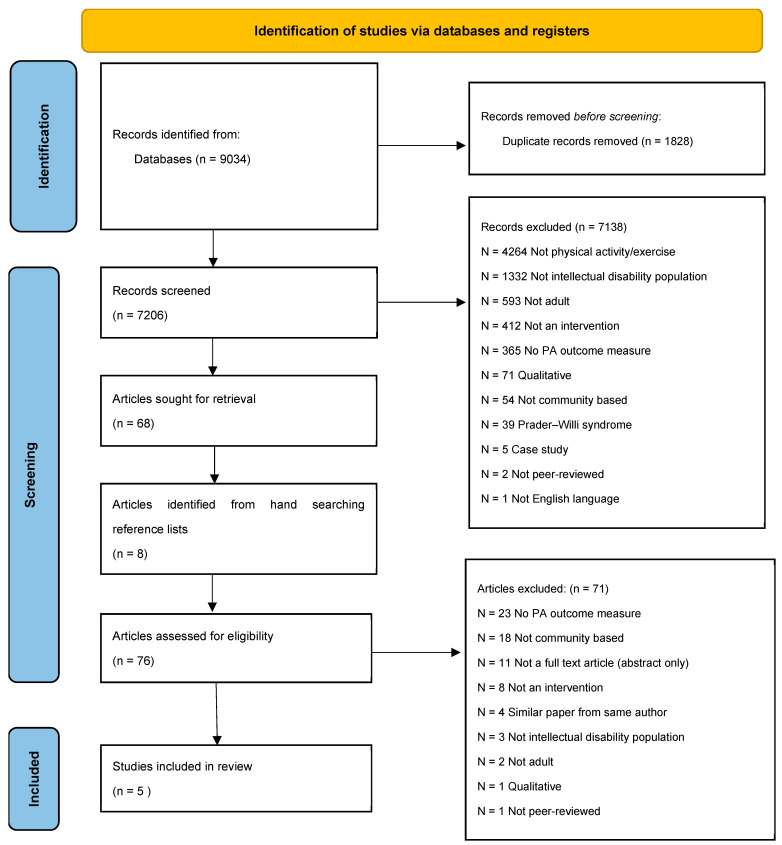
PRISMA flow diagram of systematic literature search.

**Figure 2 healthcare-13-00299-f002:**
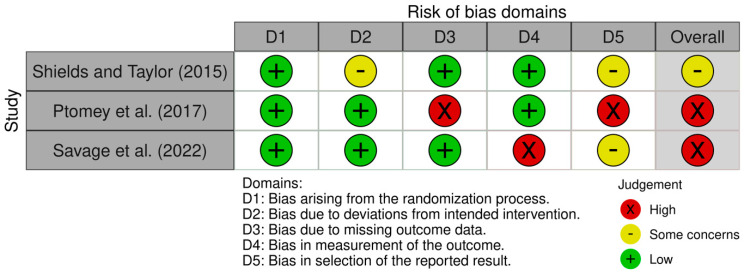
ROB2 risk of bias assessment [37,40,41].

**Figure 3 healthcare-13-00299-f003:**
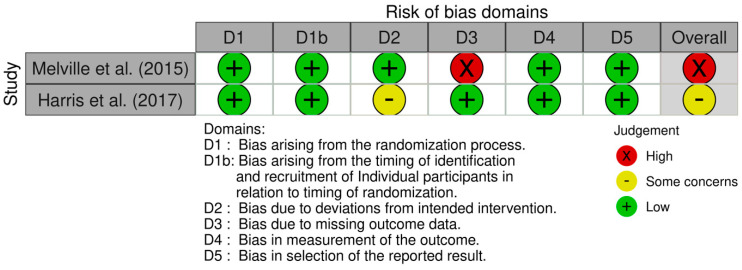
ROB 2 CRT risk of bias assessment [38,39].

**Table 3 healthcare-13-00299-t003:** CASP quality appraisal of the included studies.

	Shields and Taylor [41]	Melville et al. [39]	Ptomey et al. [37]	Savage et al. [40]	Harris et al. [38]
Are issues clearly focused?	Yes	Yes	Yes	Yes	Yes
2. Were participants randomised?	Yes	Yes	Yes	Yes	Yes
3. Were all participants properly accounted for at its conclusion?	Yes	Yes	Yes	Yes	Yes
4. Were participants blind to the intervention given? Were the investigators ‘blind’ to the intervention they were giving to participants? Were the people assessing/analysing outcomes blinded?	No	No	No	No	No
	No	No	No	No	No
	Yes	Yes	Yes	No	Yes
5. Were the groups similar at the start of the trial?	Yes	Yes	Yes	Yes	Yes
6. Aside from the experimental intervention, were the groups treated equally?	Cannot tell	Cannot tell	Cannot tell	Cannot tell	Cannot tell
7. Were the effects of the intervention reported comprehensively?	Yes	Yes	Yes	Yes	Yes
8. Was the precision of the estimate of the intervention or treatment effect reported?	Yes—partially	Yes	Yes	No	Yes
9. Do the benefits of the experimental intervention outweigh the harms and costs?	Cannot tell	Cannot tell	Cannot tell	Cannot tell	Cannot tell
10. Can the results be applied in your context?	Yes	Yes	Yes	Yes	Yes
11. Would the experimental intervention provide greater value to the people in your care than any of the existing interventions?	Cannot tell	Cannot tell	Cannot tell	Cannot tell	Cannot tell

**Table 5 healthcare-13-00299-t005:** Physical activity outcome measures and results in the included studies.

	Shields and Taylor [41]	Melville et al. [39]	Ptomey et al. [37]	Savage et al. [40]	Harris et al. [38]
Method of PA measurement	RT3 accelerometer (Stayhealthy, Inc., Monrovia, CA, USA) Pedometer (Omron, Walking Style Pro, HJ-720ITE2) Logbook data	Actigraph GT3X accelerometer (Manufacturing Technology Inc., Crestview, FL, USA) Omron Walking Style III pedometer (Omron Healthcare Inc., Hoffman Estates, IL, USA)	Actigraph GT1X accelerometer (Pensacola, FL, USA)	Fitbit Flex2	Actigraph GT3X+ accelerometers (Manufacturing Technology Inc.) International Physical Activity Questionnaire-Short (IPAQ-S)
Compliance with PA measures	Valid accelerometer data for 75% of total sample (N = 12) Valid pedometer data for 87.5% of intervention group (N = 7)	Valid accelerometer data for 80.4% of total sample (N = 82, 42 in intervention group, 40 in control group)	Baseline—N = 66% 6 months—N = 62% 18 months—N = 36%	Data presented for 100% of participants who started the intervention	Baseline N = 94% 6 months N = 76% 12 months N = 63%
PA at baseline	PA counts (VM/min) Intervention group 342.1 ± 108.8 Control group 303.4 ± 65.4 Pedometer data not reported at baseline—used only at post-intervention to corroborate logbook data	Intervention group Steps per day 4744 ± 2076 % time in PA 35.8 ± 10.4 % time in MVPA 3.2 ± 2.7 Total MET mins per week 1367.6 ± 1629.9 Control group Steps per day 4818 ± 2784 % time in PA 33.1 ± 11.3 % time in MVPA 3.3 ± 2.9 Total MET mins per week 1150.1 ± 1059.9	~15 min per day MVPA across both diet groups Valid data for 66% of sample	Intervention (n = 18) Weekly steps 47,420 ± 14,039) Control (N = 16) Weekly steps 46,277 ± 18,095	Intervention (n = 25) Light PA (% time spent/d) 21.8 ± 6.2 MVPA (% time spent/d) 4.5 ± 2.7 Control (n = 22) Light PA (% time spent/d) 22.3 ± 8.0 MVPA (% time spent/d) 4.7 ± 3.8
PA at post-intervention	Accelerometer data PA Counts (VM/min) Intervention group 344.2 ± 158.8 Control 291.0 ± 99.4 No significant between- or within-group differences in PA counts for either group Logbook data Intervention participants walked an average 175 min per week (SD = 38, range: 146–262) Pedometer data Intervention participants walked an average 147 min per week (SD = 43 min, range: 109–237).	No statistically significant changes to any outcome measures post-intervention Between-group comparison Step count 69.5 (−1054, 1193.3), *p* = 0.90, ICC = 0.51 % time in PA −1.5 (−6.1, 3.0), *p* = 0.5, ICC = 0.22 %time in MVPA 0.3 (−0.7, 1.3), *p* = 0.55, ICC = 0.42 Total MET mins per week 56.0–428.8, 540.9), *p* = 0.82, ICC = 0.02	Not reported for the 6-month data collection period Valid data for 62% of sample	Intervention (n = 18) Weekly steps 60,241 ± 4510 Control (N = 16) Weekly steps 46,377 ± 6821 Time X Group interaction *p* = 0.03	No statistically significant between-group differences for PA measures post-intervention Light PA (% time spent/d) Mean between-group difference −0.57, *p* = 0.692 MVPA (% time spent/d) Mean between-group difference 0.50, *p* = 0.434
PA at follow-up	N/A	No within-group change in the intervention group step count (adjusted difference 113.8 steps per day, 95% confidence interval −552.3 to 779.75; *p* = 0.74).	~12 min per day across both diet groups Valid data for 32% of study sample 8% of participants met MVPA goal of ≥150 min/week across the 18-month trial	N/A	No statistically significant between-group differences for PA measures at follow-up Light PA (% time spent/d) Mean between-group difference −1.71, *p* = 0.434 MVPA (% time spent/d) Mean between-group difference 0.26, *p* = 0.726

## Data Availability

The original data presented in the study will be openly available in Queen’s University Belfast PURE repository on publication.

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
