# Peer review of "A Systematic Review of Community-Based Exercise Interventions for Adults with Intellectual Disabilities"

_healthcare, 2025, doi:10.3390/healthcare13030299_

Round 1
Reviewer 1 Report
Comments and Suggestions for Authors
-
The introduction fails to clearly articulate the specific research gap this systematic review aims to address. While previous reviews are mentioned (e.g., Hassan et al. 2019, Brooker et al. 2015), the unique contribution and necessity of this new review is not explicitly stated. Additionally, several key claims about social inclusion benefits rely on outdated citations from 2004-2006, when more recent evidence should be available and cited.
The study's aims and objectives lack specificity regarding expected outcomes and measurable targets. While broad goals are stated, more precise objectives would strengthen the methodological approach. The PROSPERO registration lacks a protocol number, which limits transparency and replicability. The authors chose a very broad date range (1995-2024) without justifying why 1995 was selected as the starting point, potentially missing relevant earlier work or including outdated studies.
A notable omission is the lack of power analysis or sample size calculations, which would help readers understand if the review has sufficient statistical power to detect meaningful effects. The process for resolving disagreements between reviewers during study selection and data extraction lacks detail. The data extraction validation process is inadequately described, raising questions about the reliability of the extraction methodology.
The high heterogeneity between included studies significantly limits meaningful synthesis of findings. The small sample of only 5 included studies raises serious questions about the review's comprehensiveness and ability to draw robust conclusions. The risk of bias assessment, while present, could be more detailed and thorough. The statistical analysis of effects is limited, and effect sizes are insufficiently reported.
The discussion inadequately addresses the limited number of included studies and fails to thoroughly explore publication bias implications. Future research recommendations lack specificity and practical guidance. Some conclusions appear overstated given the limited evidence base and small number of included studies.
Multiple APA formatting issues are present throughout the manuscript. Citations are inconsistently formatted, and the reference list requires significant corrections. Tables do not follow APA guidelines for formatting and presentation. Headers lack proper APA formatting, and page numbers and running heads are missing, all of which are required elements for academic journal submission.
Each of these issues should be addressed to improve the manuscript's quality and strengthen its contribution to the field. The formatting issues, while technical in nature, reflect a lack of attention to detail that could affect the manuscript's professional presentation and credibility.
The combination of methodological weaknesses and technical formatting issues suggests significant revision is needed before this manuscript would be suitable for publication. Particular attention should be paid to strengthening the justification for the review, improving the methodological reporting, and ensuring all formatting follows current APA guidelines.
The high heterogeneity among studies and limited sample size are particularly concerning as they fundamentally affect the review's ability to draw meaningful conclusions. These core issues need to be addressed, possibly through expanded search criteria or clearer justification for the current approach.
Author Response
Dear reviewer 1,
Many thanks for your detailed feedback and advice. Please see the attached Word document for point-by-point responses to each of your comments and suggestions. We hope they are to your satisfaction.

Reviewer 2 Report
Comments and Suggestions for Authors
The article “A Systematic Review of Community-Based Exercise Interventions for Adults with Intellectual Disabilities” provides an important exploration of a topic that has significant implications for the health and well-being of adults with intellectual disabilities (ID). While it offers a comprehensive systematic review, there are notable strengths and areas for improvement that should be addressed before the article is considered for publication. The focus on community-based exercise interventions is timely, given the rising emphasis on inclusive health promotion strategies for marginalized populations. Moreover, the systematic approach aligns with rigorous research standards, and the registration in PROSPERO enhances the review's credibility. Additionally, the inclusion of objective measures such as accelerometers strengthens the methodological rigor. The authors of this paper highlights the challenges in developing effective interventions, including the lack of theoretical underpinning in some studies and inconsistent measurement approaches.
1. The small number of studies (n=5) and the narrow geographic focus (Australia, Scotland, USA) limit the generalizability of findings. Populations from Asia, Africa, and South America are notably absent. I would recommend acknowledging this limitation more explicitly and advocate for a broader geographical scope in future research.
2. Some tables in this article seem too long. I would recommend some of the split in two (e.g. 1 table splitting information to descriptive data and intervention data. Additionally in some tables there some repetitive data (e.g. table 3 and 4 first rows about Method od PA measurement are the same).
3. Some abbreviations used in the tables explained just in the text below (e.g. WC abbreviation in table 3).
4. The studies reviewed had a high risk of bias, which makes the findings less reliable. To improve, the authors should clearly explain the sources of bias and suggest ways to make future studies stronger, like using better randomization methods or ensuring those analysing the results don't know which group participants were in. Additionally, I would recommend authors to discuss how future research could better embed different theories and evaluate their impact on participant outcomes.
5. Some layout inaccuracies are noticeable in the article (e.g. long empty places in 5 page and some empty row in tables.
6. Reference list for such kind of article is appropriate. I would like to note that more than half of references (35 references) were published recently (after 2018 year).
Author Response
Dear reviewer 2,
Many thanks for your feedback and suggestions on our paper. Please find attached our responses to each of your suggestions. We hope they are to your satisfaction.

Round 2
Reviewer 1 Report
Comments and Suggestions for Authors
Authors could include more discussion of behavioral change theories and expand on implementation challenges in different settings.
While there are areas that could be further enhanced, this version represents a well-conducted systematic review that makes a valuable contribution to the field. The authors have addressed key methodological requirements and provided important insights into community-based exercise interventions for adults with intellectual disabilities. The paper is ready for publication in its current form, as the authors have made the necessary revisions to meet publication standards.
Reviewer 2 Report
Comments and Suggestions for Authors
Authors improved article according to my first-round review comments. There is still left some inaccuracies with table presentation (e.g. their split in 2 pages), but I think it could be fixed with editor.